# Time- and Dose-Dependent Effects of Ionizing Irradiation on the Membrane Expression of Hsp70 on Glioma Cells

**DOI:** 10.3390/cells9040912

**Published:** 2020-04-08

**Authors:** Helena Fellinger, Stefan Stangl, Alicia Hernandez Schnelzer, Melissa Schwab, Tommaso Di Genio, Marija Pieper, Caroline Werner, Maxim Shevtsov, Bernhard Haller, Gabriele Multhoff

**Affiliations:** 1Radiation Immuno-Oncology Group, Center for Translational Cancer Research (TranslaTUM), School of medicine, Technical University of Munich (TUM), 81675 Munich, Germany; skygallery@web.de (H.F.); Stefan.stangl@tum.de (S.S.); Alicia.hernandez@tum.de (A.H.S.); Melissa.schwab@tum.de (M.S.); tommasodigenio@hotmail.it (T.D.G.); Marija.pieper@tum.de (M.P.); c.werner@tum.de (C.W.); Maxim.shevtsov@tum.de (M.S.); 2Institute of the Russian Academy of Sciences (RAS), 194064 St. Petersburg, Russia; 3Department of Biotechnology, Pavlov First Saint Petersburg State Medical University, 197022 St. Petersburg, Russia; 4Institute of Medical Informatics, Statistics and Epidemiology, Technical University of Munich (TUM), 81675 Munich, Germany; Bernhard.haller@tum.de; 5Department of Radiation Oncology, School of Medicine, Technical University of Munich (TUM), 81675 Munich, Germany

**Keywords:** radiotherapy, glioblastoma, membrane Hsp70, dose- and time-kinetics, NK cell-based therapy

## Abstract

The major stress-inducible protein Hsp70 (HSPA1A) is overexpressed in the cytosol of many highly aggressive tumor cells including glioblastoma multiforme and presented on their plasma membrane. Depending on its intracellular or membrane localization, Hsp70 either promotes tumor growth or serves as a target for natural killer (NK) cells. The kinetics of the membrane Hsp70 (mHsp70) density on human glioma cells (U87) was studied after different irradiation doses to define the optimal therapeutic window for Hsp70-targeting NK cells. To maintain the cells in the exponential growth phase during a cultivation period of 7 days, different initial cell counts were seeded. Although cytosolic Hsp70 levels remained unchanged on days 4 and 7 after a sublethal irradiation with 2, 4 and 6 Gy, a dose of 2 Gy resulted in an upregulated mHsp70 density in U87 cells which peaked on day 4 and started to decline on day 7. Higher radiation doses (4 Gy, 6 Gy) resulted in an earlier and more rapid onset of the mHsp70 expression on days 2 and 1, respectively, followed by a decline on day 5. Membrane Hsp70 levels were higher on cells in G2/M than in G1; however, an irradiation-induced cell cycle arrest on days 4 and 7 was not associated with an increase in the mHsp70 density. Extracellular Hsp70 concentrations in the supernatant of irradiated cells were significantly higher than sham (0 Gy) irradiated cells on days 4 and 7, but not on day 1. Functionally, elevated mHsp70 densities were associated with a significantly better lysis by Hsp70-targeting NK cells. In summary, the kinetics of changes in the mHsp70 density upon irradiation on tumor cells is time- and dose-dependent.

## 1. Introduction

The therapeutic application of ionizing radiation, either alone or in combination with surgery and chemotherapy, plays a pivotal role in the treatment of solid tumors. Ionizing radiation elicits its cytotoxic activity against tumor cells by causing DNA single- and double-strand breaks, predominantly mediated and fixed by reactive oxygen species (ROS). Although the major goal of irradiation is to achieve local tumor control and a decreased dissemination by reducing the viable tumor mass, also immunostimulatory effects have been assigned to ionizing radiation [1,2]. However, it remains to be elucidated which dose and radiation regimen is optimal to induce anti-tumor immune effects. Herein, cytosolic, extracellular and plasma membrane-bound heat shock protein 70 (Hsp70) [3,4] was studied in highly aggressive glioblastoma cells as a radiation-inducible target for natural killer (NK) cells. 

Besides its intracellular chaperoning functions, which include assisting the correct folding of proteins, the assembly of nascent polypeptides, the prevention of protein aggregation and thereby controlling cellular protein homeostasis, members of the HSP70 family fulfil various cytoprotective tasks under physiological conditions and upon environmental stress [3]. An up-regulated Hsp70 expression enhances the viability of tumor cells by fostering protein damage repair and impairing apoptotic pathways [5]. By stimulating the overexpression of anti-apoptotic proteins (Bcl-2/Bcl-xL), downregulating pro-apoptotic Bax, Bcl-Xs and Bak or blocking tumor necrosis factor (TNF)-related apoptosis-inducing ligand (TRAIL)-induced apoptosis and formation of the death-inducing signaling complex with death receptors DR4 and DR5 [5], cytosolic Hsp70 promotes tumor cell survival, protects against apoptosis and promotes tumor progression [6].

A large variety of stress factors including hypoxia and reoxygenation, heavy metals, glucose deprivation and cytotoxic drugs can induce the Hsp70 expression and stimulate antigen release and danger signal expression [6,7].

In contrast to intracellular Hsp70, extracellular and mHsp70 play a pivotal role in stimulating both adaptive and innate immune responses and thereby initiating protective anti-tumor immunity [8]. By its exclusive expression on the plasma membrane of tumor cells [9], mHsp70 provides a tumor-specific target for Hsp70-specific, activated NK cells [10]. Therefore, increasing the density of mHsp70 on tumor cells by irradiation combined with a mHsp70-targeting NK cell-based immunotherapy might provide a novel strategy to improve clinical outcome and extend overall survival of patients with advanced tumors [11,12,13]. Herein, we were interested to identify the optimal radiation dose and time point for an up-regulated mHsp70 density as a target for NK cells on highly aggressive glioblastoma cells. 

## 2. Materials and Methods

### 2.1. Cell Line and Culture Conditions

The epithelial human glioblastoma cell line U87 MG (ATCC HTB-14, MGMT^-^) and the human hepatocellular carcinoma cell line HepG2 (ATCC HB-8065) were grown in complete growth medium, consisting of Dulbecco’s Eagle’s Minimum Essential Medium (DMEM) (Sigma-Aldrich, Steinheim Germany) supplemented with 10% *v*/*v* heat inactivated fetal calf serum (FCS) (Sigma-Aldrich), 1% antibiotics (10,000 IU/mL penicillin, 10 mg/mL streptomycin, Sigma-Aldrich), l-glutamine (Sigma-Aldrich), MEM non-essential amino acid solution 100× (Sigma-Aldrich) and sodium pyruvate (Sigma-Aldrich).The epithelial human cervix carcinoma cell line HeLa (ATCC CCL-2) was grown in complete growth medium, consisting of RPMI-1640 (Sigma-Aldrich, Germany) supplemented with 10% *v*/*v* heat inactivated FCS (Sigma-Aldrich), 1% antibiotics (10,000 IU/mL penicillin, 10 mg/mL streptomycin, Sigma-Aldrich), l-glutamine (Sigma-Aldrich) and sodium pyruvate (Sigma-Aldrich). After reaching confluency, adherent growing tumor cells were trypsinized for 2 min at 37 °C in trypsin ethylene diamine-tetra-acetic acid (EDTA) (Sigma-Aldrich). Single cell suspensions with different cell counts were seeded in 15 mL supplemented medium in T-75 ventilated culture flasks. Tumor cells were routinely checked for mycoplasma contamination.

### 2.2. Irradiation

Tumor cells were irradiated with a single dose of 0 (sham), 2, 4 and 6 Gy using the Gulmay RS225A irradiation machine (Gulmay Medical Ltd., Camberley, UK) at a dose rate of 0.90 Gy/min (15 mA, 200 keV) or were kept untreated. 

### 2.3. Flow Cytometry and Cell Cycle Analysis

Single cell suspensions of sham (0 Gy) irradiated and irradiated cells (0.4 × 10^6^ cells per vial) were collected at different time-points after radiation. After a washing step in phosphate-buffered saline (PBS)/10% *v*/*v* fetal calf serum (FCS), cells were incubated either with fluorescein-isothiocyanate (FITC)-conjugated mouse monoclonal antibody (mAb) specific for mHsp70 (cmHsp70.1, IgG1, multimmune GmbH, Munich, Germany) or with an isotype-matched FITC-labeled control antibody on ice in the dark for 30 min. Only viable cells (propidium iodide negative cells) were gated, and the proportion of positively stained cells and mean fluorescence intensity (mfi) values were analyzed on a FACSCalibur™ flow cytometer (BD Biosciences, Heidelberg, Germany). The mfi is a relative value of the total fluorescence intensity of cmHsp70.1-FITC antibody stained, viable cells subtracted by the intensity of the signal intensity obtained after staining of the cells with an isotype-matched IgG1-FITC control antibody. Fluorescence data were analyzed and plotted by using CellQuest software (BD Biosciences, Heidelberg, Germany). 

For a concomitant analysis of the mHsp70 expression during the cell cycle, viable cells which have been stained with cmHsp70.1 mAb were washed and fixed in 2% *w*/*v* paraformaldehyde (PFA) and then ice-cold methanol (70% *v*/*v*). After rehydration, cells were suspended in 500 µL propidium iodide/RNase staining solution (Sigma/Aldrich), incubated for 60 min at room temperature and analyzed on a FACSCalibur™ flow cytometer, as described above. 

### 2.4. Immunocytochemistry (ICC)

Cells were grown on poly-l-lysine-coated glass slides. On days 4 and 7 after irradiation with 0 (sham) and 4 Gy, cells were stained with FITC-labeled cmHsp70.1 monoclonal antibody (multimmune GmbH, Munich) on ice for 30 min. After antibody incubation, cells were washed in ice-cold PBS and fixed in 3.7% *w*/*v* PFA in PBS (pH 7.4). Nuclei were counterstained with 4’,6-diamidino-2-phenylindole (DAPI). Fluorescence images were taken with an AxioImager M2 microscope, equipped with a 100× oil immersion objective (Carl Zeiss Microscope, Jena, Germany) at a resolution of 2048 × 2048 pixels. To avoid potential cross-interferences of the different fluorophores, images for FITC and DAPI were acquired using a sequential image recording mode. 

### 2.5. Hsp70 lipELISA

Levels of extracellular Hsp70 in the supernatant of sham irradiated and irradiated U87 cells on days 1, 4 and 7 were determined using the lipHsp70 ELISA, as described elsewhere [14]. The measured values of extracellular Hsp70 were normalized to 1 × 10^6^ viable tumor cells. 

### 2.6. Europium Assay 

The lytic activity of human NK cells cultured in RPMI-1640 medium supplemented with 10% *v*/*v* heat inactivated fetal calf serum (FCS), 1% antibiotics (10,000 IU/mL penicillin, 10 mg/mL streptomycin), l-glutamine, sodium pyruvate and Hsp70 peptide TKD (2 µg/mL) and IL-2 (100 IU/mL) at a cell density of 5 × 10^6^ peripheral blood lymphocytes (PBL) for 4 days against sham (0 Gy) and 4 Gy irradiated tumor cells was determined at different effector-to-target (E:T) ratios ranging from 50:1 to 3:1 using a standard 3.5 h Europium assay. The tumor cells were washed twice with fresh medium before the assay starts to exclude the presence of extracellular Hsp70 during the assay. Mean values of an experiment in triplicates are shown. The specific lysis was calculated according to Equation (1): (1)$$Specific\;Lysis=\frac{Experimental\;Release-Spontaneous\;Release}{Maximum\;Release-Spontaneous\;Release}$$

### 2.7. Statistical Analysis

Each sample was measured in at least three independent experiments. Means and standard deviations are presented for quantitative data. Analysis of variance (ANOVA) was used for comparisons of group means. If the global null hypothesis of all group means being equal could be rejected on a significance level of α = 5%, pairwise group comparisons were conducted by *t*-tests. For these pairwise comparisons, standard deviations were pooled over all groups, and Bonferroni correction was applied to account for multiple testing. A two-way ANOVA was used to assess the impact of radiation dose on the specific lysis at different E:T ratios by fitting a model including main effects and their interaction term to the data. A significance level of 5% was used.

## 3. Results

### 3.1. Membrane Hsp70 Expression Remains Stable During a Culture Period of 7 Days

To maintain cells in the exponential growth phase during a culture period of 7 days, different tumor cell counts were seeded. For a culture period of 1–4 days 0.25 × 10^6^ cells, for 5 days 0.125 × 10^6^ cells, for 6 days 0.06 × 10^6^ cells and for 7 days 0.01 × 10^6^ cells were seeded on day 0. Representative data for mHsp70 expression on U87 cells during a period of 7 days after sham irradiation (0 Gy) are summarized in Figure 1. No significant differences in the percentage of positively stained cells (Figure 1A; F(6,14) = 0.526, *p* = 0.780) and mean fluorescence intensity (mfi) (Figure 1B; F(6,14) = 2.202, *p* = 0.105) were detected over a 7 day culture period. The slightly lower mHsp70 density on day 0 could be attributed to the very short incubation period of 5 h after trypsin treatment and seeding. 

### 3.2. Irradiation Induces an Increase in the Mhsp70 Density on Different Tumor Cell Lines

Although the percentage of mHsp70-positive cells remained unaltered on U87 (Figure 2A), HeLa (Figure 2B) and HepG2 (Figure 2C) cells on day 1 after sham (0 Gy), 4 and 6 Gy irradiation, the mHsp70 density differed significantly upon radiation. A radiation dose of 6 Gy nearly doubled the mean fluorescence intensity (mfi) in all three tumor cell lines. In U87 cells the mfi increased from 38.9 ± 15.8 (sham) to 47.1 ± 16.0 after irradiation with 4 Gy and to 83.5 ± 14.2 after irradiation with 6 Gy, on day 1 (Figure 2D; F(2,6) = 4.77, *p* = 0.058). In HeLa cells the mfi increased from 9.2 ± 1.9 (sham) to 15.7 ± 0.8 (*p* = 0.04) and 20.6 ± 6.8 (*p* = 0.03) after irradiation with 4 and 6 Gy, respectively (Figure 2E), and in HepG2 the mfi increased from 13.9 ± 2.7 to 22.0 ± 2.6 (*p* = 0.04) after 4 Gy and to 22.3 ± 4.6 (Figure 2F) after 6 Gy, on day 1. The mHsp70 density on sham treated and completely untreated control cells was identical (data not shown). Cell viability of all tumor cell lines was not affected by radiation doses ranging from 2 to 6 Gy.

### 3.3. Irradiation-Induced Upregulation of mHsp70 Is Time-Dependent

A comparison of the mHsp70 densities on days 4 and 7 after irradiation with 4 and 6 Gy revealed that mfi values on day 4 remained significantly upregulated, but dropped on day 7 after radiation in HeLa (Figure 3A) and U87 (Figure 3B) cells. The mfi of sham (0 Gy) treated HeLa cells was 7.9 ± 1.3 compared to 14.8 ± 3.0 and 18.6 ± 3.1 after radiation with 4 and 6 Gy on day 4 (Figure 3A; *p* = 0.05). The mfi of sham (0 Gy) treated U87 cells was 41.9 ± 0.8 compared to 79.3 ± 12.8 and 75.1 ± 7.7 after radiation with 4 Gy (*p* = 0.02) and 6 Gy (*p* = 0.03) on day 4 (Figure 3B; F(2,6) = 11.24, *p* = 0.009). Significant differences were observed between sham and 4 Gy and between sham and 6 Gy irradiated cells, whereas no statistically significance was observed between 4 and 6 Gy irradiated HeLa (Figure 3A) and U87 cells (Figure 3B). On day 7 after irradiation, the mfi of sham treated HeLa cells remained at 9.2 ± 1.4, but that of 4 and 6 Gy irradiated cells dropped to 12.4 ± 1.0 and 17.5 ± 1.7, respectively, on day 7 (Figure 3A). In line with these results, the mfi of sham treated U87 cells remained at 38.73 ± 1.41, but that of 4 and 6 Gy irradiated cells dropped to 37.60 ± 7.24 and 40.02 ± 5.71, respectively, on day 7 (Figure 3B; F(2,6) = 0.1003, *p* = 091). No significant differences in the mfi were observed in the differently irradiated U87 cells on day 7 (Figure 3B). The upregulated mHsp70 density on day 4 and the downregulation of mHsp70 on day 7 after irradiation with 4 Gy was confirmed by immunocytochemistry (ICC) (Figure 3C). In contrast to the mHsp70 expression, the cytosolic Hsp70 content in U87 cells remained unaltered upon irradiation with 2, 4 and also 6 Gy, as determined by Western blot (Figure 3D) and intracellular Hsp70 staining on days 4 and 7 (Figure 3E). Similar results were observed with HeLa cells (data not shown).

### 3.4. Irradiation-Induced Effects on the mHsp70 Density Are Dose- and Time-Dependent

The previous results indicated clear differences in the mHsp70 density on U87 cells that depend on the recovery time after irradiation. Figure 4 summarizes dose- and time-dependent effects on the mHsp70 density. Although mfi values for sham (0 Gy) treated cells did not differ significantly over a period of 7 days when different initial cell counts were seeded (Figure 1B), an exposure to 2 Gy induced a progressive slow increase in mfi values form day 0 up to a maximum value of 67.4 ± 16.7 on day 5 that decreased to 39.3 ± 4.3 on day 7 (Figure 4A; F(6,14) = 3.8952, *p* = 0.017). Cells that have been irradiated with a higher dose (4 Gy) reached the maximum mfi value (79.3 ± 12.8; *p* ≤ 0.05) by day 4 (Figure 4B; F(6,14) = 3.916, *p* = 0.016); however, the drop in the mfi values thereafter occurred more rapidly than after irradiation with 2 Gy. Following an irradiation with 6 Gy, mHsp70 expression density was significantly increased already on day 1 compared to sham (0 Gy) irradiated cells (23.3 ± 3.7 versus 83.6 ± 14.2; *p* ≤ 0.05), persisted for 3 days and dropped on day 5 (Figure 4C; F(6,14) = 10.70, *p* = 0.0002). In Figure 4A–C, mfi values differing significantly (*p* ≤ 0.05) from the value on day 0 at each individual irradiation dose (2, 4, 6 Gy) using a pairwise comparison *t*-test with pooled SD marked with an asterisk. A comparison of all time points (day 0 to day 7) and all radiation doses (0, 2, 4, 6 Gy) reflects the different dose- and time-kinetics of the mHsp70 densities in U87 cells (Figure 4D). 

### 3.5. Irradiation-Induced Cell Cycle Arrest and Its Impact on the mHsp70 Density

On day 4 (F(3,24) = 64.83, *p* < 0.001) and on day 7 (F(3,24) = 4.232, *p* = 0.016) after irradiation with 2, 4 and 6 Gy, a dose-dependent cell cycle arrest in G2/M was observed in U87 cells (Figure 5A). A concomitant analysis of the mHsp70 density with the cell cycle revealed higher mHsp70 densities in G2/M compared to G1 in sham (0 Gy) and irradiated (2, 4, 6 Gy) tumor cells (Figure 5B). However, the mHsp70 density appeared not to be associated with the irradiation-induced G2/M arrest since a higher proportion of cells in G2/M on day 7 was not associated with a further increase in the mHsp70 density. An assessment of extracellular Hsp70 concentrations normalized to 1 × 10^6^ viable tumor cells showed significantly increased values on days 4 and 7 after an irradiation with 4 and 6 Gy compared to sham irradiated cells (*p* ≤ 0.05), whereas on day 1 no significant increase in extracellular Hsp70 in the cell culture supernatant was detected (Figure 5C). 

### 3.6. Increased mHsp70 Expression Density Is Associated with an Increased Sensitivity to Lysis Mediated by TKD/IL-2-Activated NK Cells

The density of mHsp70 expression plays a crucial role in the recognition of tumor cells by Hsp70-peptide TKD/IL-2-activated NK cells, but not by cytotoxic CD8-positive T lymphocytes, as previously demonstrated [15]. Therefore, the cytolytic activity of Hsp70-targeting NK cells against sham (0 Gy) and irradiated (4 Gy) tumor cells was tested. The increase in the mHsp70 density of tumor cells on day 4 after irradiation that were used in the cytotoxicity assay is illustrated in Figure 6A. The lysis of irradiated (4 Gy) tumor cells by TKD/IL-2-activated NK cells is significantly higher than that of sham (0 Gy) irradiated cells at E:T ratios ranging from 50:1 to 3:1 (Figure 6B; F(1,4) = 8.11, *p =* 0.00004).

## 4. Discussion

Compared to normal cells, highly aggressive tumor cells frequently exhibit higher cytosolic Hsp70 levels, which are further upregulated upon a variety of different environmental stress factors [7]. Intracellularly, Hsp70 ensures correct protein folding and transport [3] and interferes with both intrinsic and extrinsic apoptosis pathways to avoid cell death and thereby foster tumor cell survival [5]. Apart from its cytosolic localization, Hsp70 can be actively released in lipid microvesicles, such as exosomes [13,16,17,18], and Hsp70 is also expressed on the plasma membrane of tumor cells [19]. While cytosolic Hsp70 is mainly responsible for maintaining protein homeostasis and protection against programmed cell death [3,5], mHsp70 mediates dual functions: on the one hand it stabilizes lysosomal and plasma membranes and thereby also can prevent apoptosis [20,21], on the other hand, it provides a tumor-specific target for NK cells that have been pre-activated with Hsp70 plus low-dose IL-2 [8,15]. The specific membrane localization of Hsp70 on tumor cells can be explained by an interaction of Hsp70 with tumor-specific lipid compounds such as globoyltriaosylceramide Gb3 [22]. In vitro lipid copellation assays revealed that recombinant Hsp70 specifically interacts with the tumor-specific lipid raft component globoyltriaosylceramid Gb3 [22] or with the non-raft lipid compound phosphatidylserine (PS), which translocates to the outer plasma membrane leaflet upon environmental stress [23]. Since mHsp70 has been found to serve as a target for Hsp70-peptide TKD/IL-2-activated NK cells in vitro [24] and in vivo [25], time- and dose-dependent effects of irradiation with respect to the mHsp70 density on different human tumor cell lines were studied. The relevance of mHsp70 as a target for C-type lectin receptor CD94-positive NK cells [14] has previously been demonstrated in isogenic tumor cell systems which express differential levels of mHsp70, but identical cytosolic Hsp70 levels, and by Hsp70 antibody blocking studies [21]. A better understanding of the mHsp70 kinetics following irradiation might help to identify an optimal therapeutic window for combination therapies consisting of ionizing irradiation and Hsp70-targeting NK cell-based immune therapies. 

Glioblastoma multiforme (GBM) the most common malignant neoplasia of the brain in adults is associated with a high mortality. Despite multimodal treatment strategies consisting of surgery, radiotherapy and systemic chemotherapy using temozolomide [26], the median survival remains poor (15–18 months) and recurrence rates are high. Its intratumoral heterogeneity, concerning genetic alterations and morphology, as well as the infiltrative growth and diffuse dissemination into the brain parenchyma often limit a successful treatment of GBM. Therefore, there is a high medical need for innovative treatment strategies for GBM. The mHsp70 density as a potential target for NK cells was measured in the highly radiation-resistant human glioblastoma cell line U87 and two other tumor cell lines at different time points after irradiation with doses ranging from 2 to 6 Gy. In a previous study, we demonstrated that a high mHsp70 expression was also found on primary glioblastoma cells without isocitrate-dehydrogenase 1 (IDH-1) mutation, whereas secondary glioblastoma with IDH-1 mutation and low-grade anaplastic gliomas exhibited a lower mHsp70 density. The O6-methylguanine DNA methyltransferase (MGMT) status appeared to be not associated with the mHsp70 expression [27]. Regarding these results, we hypothesize that the MGMT-negative glioblastoma cell line U87 might serve as a model for highly aggressive, mHsp70-positive primary glioblastomas without IDH-1 mutation. 

To maintain cells in the exponential growth phase, tumor cells were seeded at different cell densities, which were adapted to the different culture periods. In sham as well as non-treated tumor cells, the proportion of mHsp70-positive cells remained at nearly 100% with a relatively low density over the whole culture period of 7 days. These findings are in line with previous results obtained with other tumor types such as epithelial tumor cells of the head and neck area [28].

Environmental stress, including irradiation, can cause an up-regulation of the mHsp70 density on tumor cells [10,20]. Herein, we have shown for the first time that the kinetics of the mHsp70 expression is dependent on the irradiation dose, but not on the cytosolic Hsp70 levels. While the maximum mHsp70 expression is reached only on day 5 after an irradiation with 2 Gy, the peak mHsp70 expression is detected already on day 1 after irradiation with 6 Gy. This means that low-dose irradiation might be associated with a later onset of the mHsp70 expression compared to high-dose irradiation. These findings are in accordance to results of Diller et al. [29] who showed that the kinetics of the heat shock protein synthesis are transient and proportional to the applied stress. 

Synthesis and expression of Hsp70 is often dependent on dose- and time-dependent factors; therefore, cytoprotective repair mechanisms that depend on the anti-apoptotic molecular chaperone Hsp70 in U87 cells are also related to the applied stress [29]. Our findings indicate that a high, but yet non-lethal, irradiation dose of 6 Gy induces an early onset of the mHsp70 expression after 1 day with a sustained overexpression that persists for 3 days, whereas lower irradiation doses induce a slower up-regulation in U87 cells. 

In an effort to define the optimal timing for the application of ex vivo stimulated mHsp70-targeting NK cells [24] after radiotherapy, it is important to determine the maximum density of mHsp70 on tumor cells. In the case of U87 cells, an irradiation dose between 4 and 6 Gy might be optimal to achieve a fast and long-lasting upregulation of the mHsp70 expression in vitro. However, future preclinical studies using fractionated irradiation protocols are needed to define the optimal irradiation dose that is suitable for an in vivo application of mHsp70-targeting immune cell-based therapies. Interestingly, in highly radiation-resistant U87 cells [29], none of the applied radiation doses caused an upregulation in intracellular Hsp70 levels. This finding might suggest that anti-apoptotic pathways that are predominantly regulated by cytosolic Hsp70 might not be initiated in U87 cells up to a radiation dose of 6 Gy. Furthermore, these data indicate that the radiation-induced translocation of Hsp70 from the cytosol to the plasma membrane is not dependent on *de novo* Hsp70 synthesis. An ER–Golgi pathway could also be ruled out for the translocation of Hsp70 from the cytosol to the plasma membrane, since inhibitors of classical protein transport pathways like monensin, brefeldin A, tunicamycin and thapsigargin neither affected mHsp70 expression nor exosomal release [30]. It is more likely that non-classical vesicular transport pathways [29] that appear to be inducible by non-lethal irradiation might enable mHsp70 translocation and release, since we observed an increase in extracellular Hsp70 on days 4 and 7 after irradiation with 4 and 6 Gy. Moreover, a radiation-induced cell cycle arrest appeared to be not associated with mHsp70 transport.

There is strong evidence that lethal stress which drastically harms tumor cell survival cannot be compensated by cytoprotective repair mechanisms that include the cytosolic overexpression of Hsp70 [31,32]. After lethal stress, Hsp70 can be externalized by dying tumor cells as a danger-associated molecular pattern (DAMP) with the capacity to initiate anti-tumor immune responses [1,2,33]. Hsp70-chaperoned tumor peptides can induce CD8-positive T cell mediated immune responses after cross-presentation of immunogenic tumor antigens on MHC class I molecules [34], whereas peptide-free Hsp70 in the context of pro-inflammatory cytokines, such as IL-2, can augment the cytolytic and migratory capacity of NK cells that recognize and kill remaining therapy-resistant mHsp70-positive tumor cells [16,21]. Herein, we could show that upon sub-lethal irradiation with 4 and 6 Gy, extracellular Hsp70 concentrations started to increase on days 4 and 7. As to whether elevated extracellular Hsp70 levels are able to further stimulate NK cell mediated immunity in vivo remains to be determined in preclinical models. 

In summary, the applied radiation dose- and time-kinetics play a critical role in optimizing radiation-induced effects by increasing the density of mHsp70 expression on the surface of surviving residual tumor cells as a target for immune cells. Membrane-bound Hsp70 plays a crucial role in stimulating the innate and adaptive immune system as it is selectively expressed on the cell surface of tumor, but not normal cells [35], and therefore serves as a tumor-specific recognition structure [13]. We could show that Hsp70-targeting NK cells are capable to specifically recognize and kill tumor cells presenting Hsp70 on their plasma membrane most likely via granzyme B mediated apoptosis [36].

## 5. Conclusions

In contrast to intracellular Hsp70, extracellular and mHsp70 play key roles in stimulating both adaptive and innate immune responses and thereby might provide protective anti-tumor immunity [8]. By its exclusive expression on the surface of tumor cells, but not normal cells [9], mHsp70 serves as a tumor-specific target for activated NK cells [10]. In this study we present evidence that the radiation dose plays a pivotal role on the kinetics of the mHsp70 density of human glioblastoma cells. Low-dose irradiation is associated with a later onset of the mHsp70 expression compared to higher irradiation doses. Therefore, hypofractionated irradiation schemes with higher doses might be beneficial for generating an extended therapeutic window for mHsp70-targeting immunotherapies.

## Figures and Tables

**Figure 1 cells-09-00912-f001:**
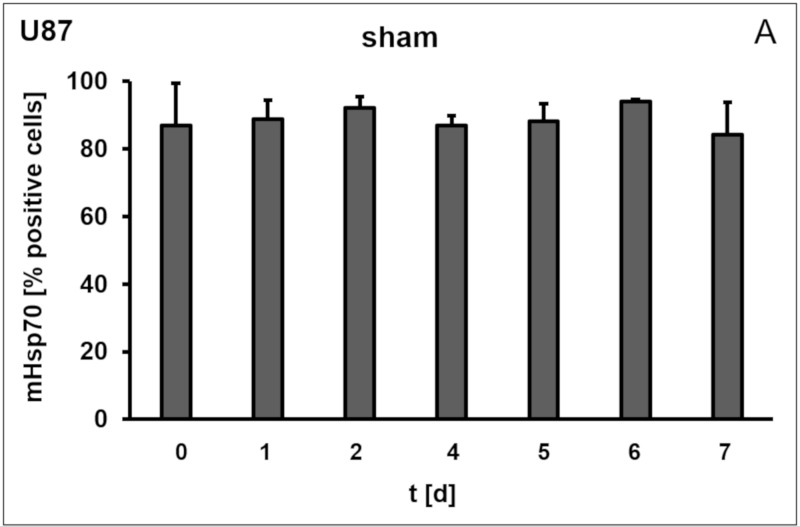
Percentage (**A**) and mean fluorescence intensity (mfi) (**B**) of mHsp70-positive U87 glioblastoma cells after sham irradiation (0 Gy) on days 0, 1, 2, 4, 5, 6 and 7. Bars represent the mean value and the corresponding standard deviation (SD) of *n* = 3 independent experiments. ANOVA was used to compare data across all days.

**Figure 2 cells-09-00912-f002:**
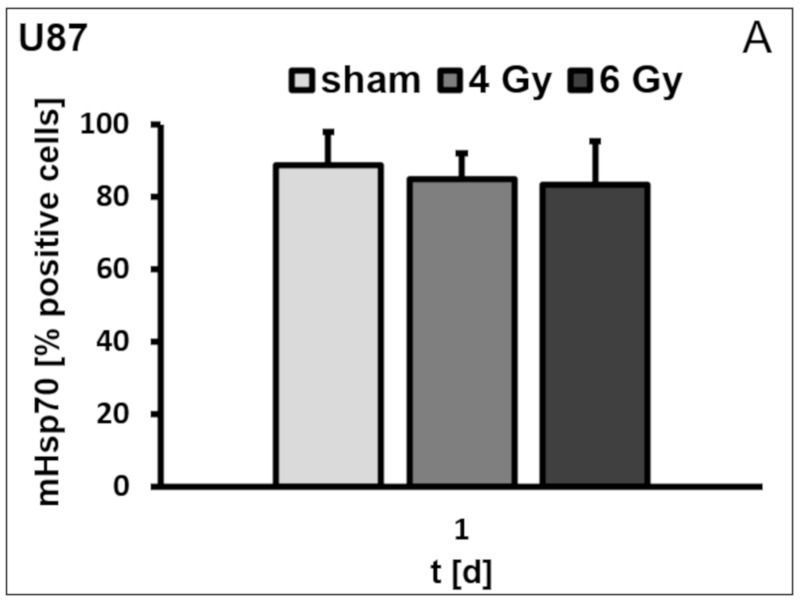
Percentage of mHsp70-positive U87 (**A**), HeLa (**B**) and HepG2 (**C**) cells on day 1 after irradiation with 0 (sham), 4 and 6 Gy. Mean fluorescence intensity (mfi) of mHsp70 expression on U87 (**D**), HeLa (**E**) and HepG2 (**F**) cells on day 1 after irradiation with 0 (sham), 4 and 6 Gy. Bars represent the mean value and the corresponding standard deviation (SD) of *n* = 3 (U87), *n* = 4 (HeLa), and *n* = 3 (HepG2) independent experiments. For comparison of different irradiation doses, ANOVA has been conducted followed by pairwise *t*-tests (with Bonferroni correction). Significance (sham vs. different irradiation doses on day 1, after Bonferroni correction): * *p* ≤ 0.05.

**Figure 3 cells-09-00912-f003:**
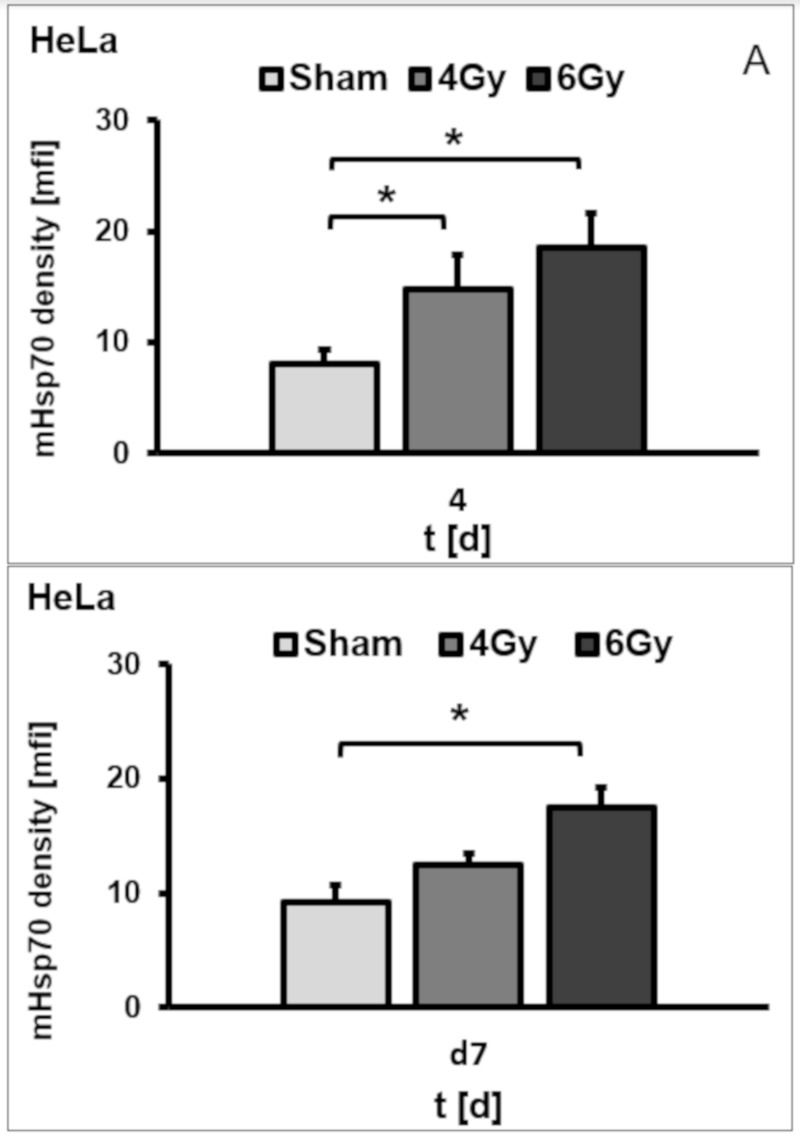
Mean fluorescence intensity (mfi) of mHsp70 expression in HeLa (**A**) and U87 (**B**) cells after sham (0 Gy), 4 and 6 Gy irradiation on days 4 and 7. Bars represent the mean values and the corresponding standard deviation (SD) of n = 3 independent experiments. Significance (sham vs. different irradiation doses on day 4): * *p* ≤ 0.05 (after Bonferroni correction). Comparison of different doses (sham 0, 4, 6 Gy) were performed by pairwise *t*-tests with pooled SD and Bonferroni correction for multiple testing. An upregulation of the mHsp70 expression density upon irradiation with 4 Gy on day 4 and a downregulation of the mHsp70 expression on day 7 shown by immunocytochemistry (ICC) in U87 cells (**C**). Cytosolic Hsp70 expression was determined by Western blot analysis (**D**) and intracellular Hsp70 staining (**E**) in U87 cells after sham (0 Gy), 4 and 6 Gy irradiation on days 4 and 7. β-actin staining served as a loading control. Data show one representative experiment out of two independent experiments with similar results.

**Figure 4 cells-09-00912-f004:**
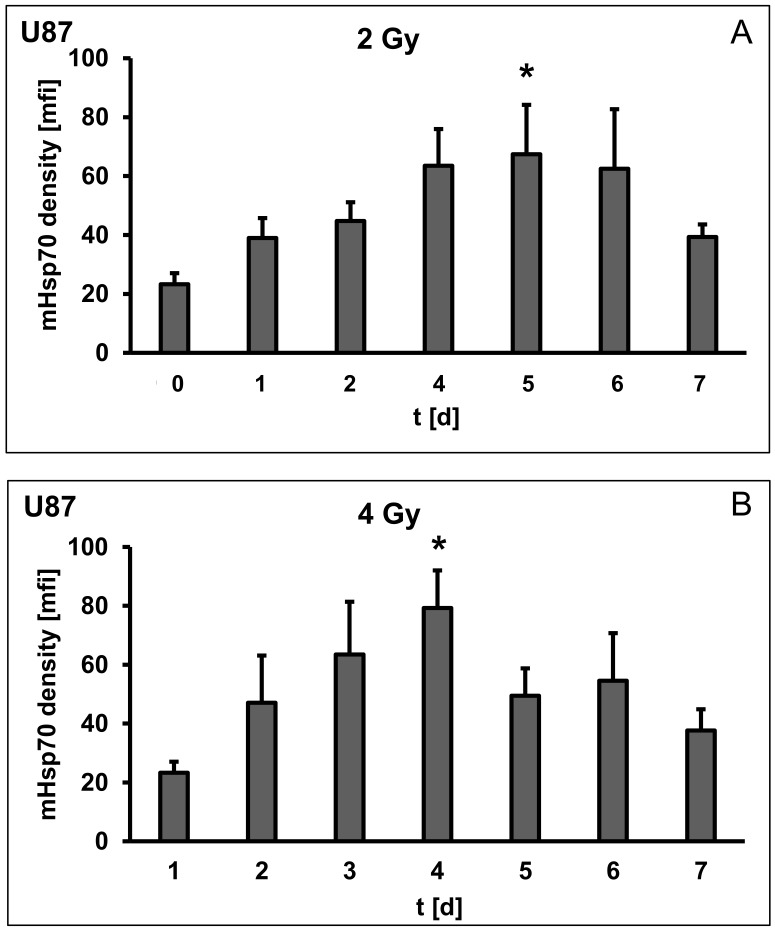
Dose- and time-dependent radiation effects on the mean fluorescence intensity (mfi) of mHsp70 on U87 human glioblastoma cells over 7 days with different radiation doses (2 Gy (**A**), 4 Gy (**B**), 6 Gy (**C**)). Bars represent the mean values and the corresponding standard deviation (SD) of *n* = 3 independent experiments. Significance (comparison of one dose on different days): * *p*
*≤* 0.05. For comparison of different days at a single dose ANOVA has been conducted. Comparisons to sham (0 Gy) treated cells were performed by pairwise *t*-tests with pooled SD and Bonferroni correction for multiple testing. (**D**) Summary of the kinetics of the mHsp70 density after sham (0 Gy), 2, 4, 6 Gy) over a period of 7 days. For a better clarity, statistical significance was only shown in Figure 4A–C, but not in Figure 4D.

**Figure 5 cells-09-00912-f005:**
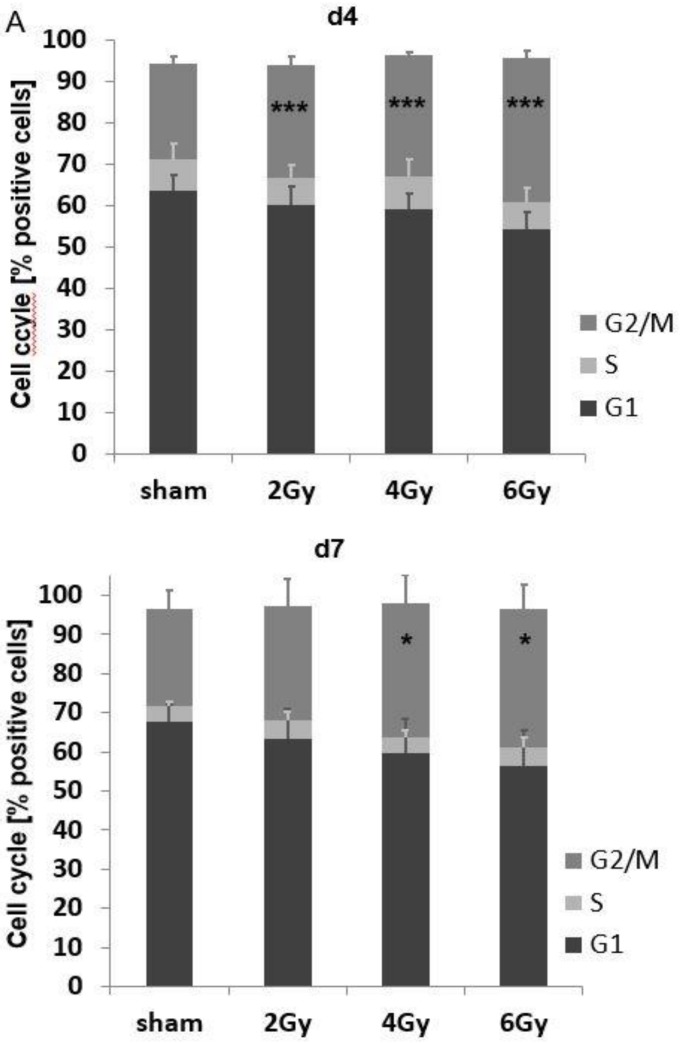
Comparative analysis of the cell cycle (G0, S, G2/M) in U87 cells after sham (0 Gy), 2, 4 and 6 Gy irradiation on days 4 and 7: ** p* ≤ 0.05; **** p* ≤ 0.001 (**A**). For comparison of different cell cycle phases ANOVA has been used. Differences in the G2/M phase were analyzed by pairwise t-tests with pooled SD and Bonferroni correction for multiple testing. Comparative analysis of the mHsp70 expression in the different cell cycle phases (G0, S, G2/M) in U87 cells on days 4 and 7 after sham (0 Gy), 2, 4 and 6 Gy irradiation (**B**). Comparative analysis of extracellular Hsp70 concentrations in the supernatant of U87 cells on days 1, 4 and 7 after sham (0 Gy), 2, 4 and 6 Gy irradiation (**C**). The data were normalized to a defined number of 1 × 10^6^ cells viable cells. Data represent the mean values and the corresponding standard deviation (SD) of *n* = 3 independent experiments. Significance (sham vs. all different irradiation doses): ** p* ≤ 0.05. For comparisons of extracellular Hsp70 levels, pairwise *t*-tests with pooled SD and Bonferroni correction were performed.

**Figure 6 cells-09-00912-f006:**
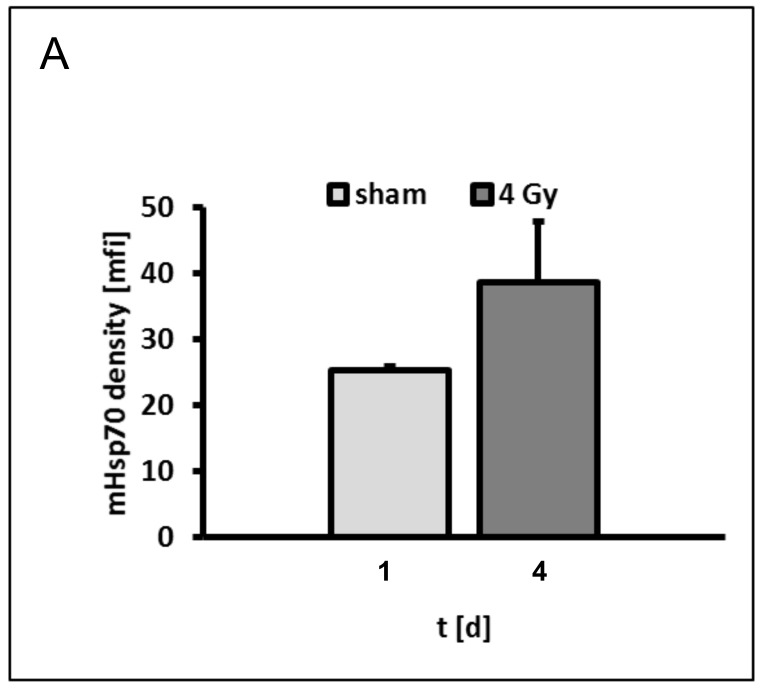
Comparative analysis of the cytolytic activity of TKD/IL-2-activated natural killer (NK) cells targeting mHsp70-positive tumor cells. Sham (0 Gy) and 4 Gy irradiated tumor cells on day 4 were used as target cells. (**A**) mHsp70 mean fluorescence intensity (mfi) on sham (0 Gy) and 4 Gy irradiated tumor target cells. The results represent mean values and the corresponding standard deviation (SD) of *n* = 2 independent experiments. The Welch two sample *t*-test was used. (**B**) Lytic activity of NK cells against sham (0 Gy) and 4 Gy irradiated tumor target cells. The effector to target (E:T) ratios range from 50:1 to 3:1. The results represent mean values and the corresponding standard deviation (SD) of *n* = 2 independent experiments in triplicate. Significance (lysis of sham vs. 4 Gy irradiated target cells): **** p* ≤ 0.001. Two-way ANOVA were used to compare lysis at different E:T ratios against tumor cells after different irradiation (sham (0 Gy) vs. 4 Gy).

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
