# Peer review of "Time- and Dose-Dependent Effects of Ionizing Irradiation on the Membrane Expression of Hsp70 on Glioma Cells"

_cells, 2020, doi:10.3390/cells9040912_

Round 1

Reviewer 1 Report

All my concerns have been addressed. Please check out the error bars in each figure. Some of them look very wired to me. 

Author Response

Thank you once again for helpful suggestions. The error bars have been unified as recommended.

Reviewer 2 Report

Title: Time- and dose-dependent effects of ionizing irradiation on the membrane expression of Hsp70 on glioma cells

Authors: Helena Fellinger et al.

In this manuscript, the authors demonstrated the radiotherapy dose and time effects on the expression of Hsp70, a stress inducible protein, in a human glioma cell line (U87), and in two carcinoma cell lines (HeLa and HepG2). The findings regarding U87 cell line are of importance to design further treatment, namely using Hsp70-targeting NK cells.

The manuscript is well written. All needed informations are found. The results are step by step presented with a perfect scientific logic. Experiments are well conducted and presented. In the revised version, authors add informations on two cell lines: HeLa and HepG2. This is only an indirect demonstration of how Hsp-70 could behave in glioblastomas of distinct sub-classes, since HeLa and HepG2 are carcinoma-derived cell lines. Meanwhile it should not restrain manuscript publication. Also addition of Hsp-70 immunohistochemistry, at different times after irradiation, gives an additional improvement of the data.

In the figures, some are labeled with d and the day of treatment. Others only include the day of treatment. Similar labeling should be used throughout figures.

To conclude: a nice manuscript interesting for the entire neuro-oncology community and not only for radiotherapist.

Author Response

Thank you for helpful suggestions. Thee labels of the Figures have been unified in a new version of the Figures.

Reviewer 3 Report

The authors have addressed all the concerns raised.

Minor edits with respect to English (sentencing, grammar, etc) is required.

Once those are corrected by the editorial team - the paper should be good for publication. No need of second round of review.

Author Response

Thank you for helpful suggestions. The Ms was proof-read by a native English speaker. We are also thankful for corrections by the editorial board.

This manuscript is a resubmission of an earlier submission. The following is a list of the peer review reports and author responses from that submission.

Round 1

Reviewer 1 Report

Manuscript: Cells-688062

Time- and dose-dependent effects of ionizing irradiation on the membrane expression of Hsp70 on glioma cells

In this in vitro study, the authors used three cell lines (U87, HeLa and HepG2) to show the radiation response of membrane-bound Hsp70 expression. Particularly, in ref to U87, authors conclude that low dose (2 Gy) irradiation upregulates mHsp70 with a peak at day four and decline by day 7. Higher radiation doses (4-6 Gy) showed an early peak (day 2 and 1) and decline by day 5. In the end, higher Hsp70 upregulation leads to increased lysis by NK cells. The authors used pretty standard techniques (cell density measurements, flow cytometry) to conduct the experiments.

1) Authors strongly propose that compared to the cytosolic isoform of Hsp70, membrane-bound isoform may promote tumor cell survival and progression. However, to clearly differentiate roles of cytosolic vs mHsp70 following exposure to low to high doses of irradiation, both isoforms should be studied to show tumorigenic (or radiation-specific) induction of mHsp70 (not cytosolic Hsp70).

2) Apart from flow cytometry, IHC or ICC staining of cells exposed to 0, 2-6 Gy irradiation at various time points should be able to convince the reviewer that specifically mHsp70 upregulated by irradiation.

3) Is there any correlation between cell cycle arrest post-irradiation and mHsp70 expression? Assessment of cell cycle phases in relation to radiation dose and time post-irradiation should be able to explain the mHsp70 induction dynamics better?

4) What factor, in particular, induces mHsp70 expression? Radiation-induced oxidative stress, cell cycle arrest or elevated apoptosis? This study could not explain these points.

5) Radiation exposure may lead to cell death and release of cellular contents containing both cytosolic and membrane-bound Hsp70 in the media. Secondly, as the authors stated, Hsp70 can also be packaged into exosomes and this could be either cytosolic or membrane-bound or both isoforms. This is a critical point given the inability to differentiate either isoform in the media and/or via flow cytometry analysis.

Moreover, such release of Hsp70 could also activate NK cells and therefore cell lysis ability may not be related to either membrane-bound or cytosolic overexpression of Hsp70.

6) Statistical approach lack some depth

IN GENERAL, (for all data sets), authors should show if overall ANOVA is significant and provide P-value (F-stats details). Secondly, for data set involving multiple groups, Bonferroni’s or equivalent post hoc correction should be provided to derive statistical significance. A detailed stat should be included in the Results and a brief statement should be provided in each figure. Two away ANOVA should be provided to derive interaction of Con, three drugs and/or cell lines throughout the manuscript.

Overall, this is a correlative study and authors need to answer all major concerns and suggestions to improve the manuscript and bring it to the publication level.

Reviewer 2 Report

Title: Time- and dose-dependent effects of ionizing irradiation on the membrane expression of Hsp70 on glioma cells

Authors: Helena Fellinger et al.

In this manuscript, the authors demonstrated the radiotherapy treatment dose and time effects on the expression of Hsp70, a stress inducible protein, in a human glioma cell line (U87).

The manuscript is well written. All needed informations are found. The results are step by step presented with a perfect scientific logic. Experiments are well conducted and presented.

The main concern of this work is that it has only been performed using a single cell line. Diffuse glioma is not a single disease but regrouped IDH1/2 non-mutated and mutated gliomas. Amongst the former, IDH wild-type glioblastoma divide in different sub-types according to the TCGA classification, or to the DKFZ methylation classes. Therefore, various cell lines should be studied and compared.

Reviewer 3 Report

In this MS, Fellinger H et al. examined the effects of different dose of irradiation on the expression of HSP70 in multiple cancer cells. They found that both low and higher doses irradiation can induce the expression of HSP70 but with distinct pattern. Finally, they showed that the levels of HSP70 functionally related to the lysis mediated by HSP70-targeting NK cells. Although this study provides a novel insight about the effects of irradiation on HSP70 expression, the data are preliminary and more experiments are obviously needed to support some conclusion.

Major concerns:

How about the expression of HSP70 during culture in Hela and HepG2 cells? It looks like that Figure 2 and 3 are some kinds of overlapping with Figure 4. If so, please try to combine them. The data of figure cannot support their conclusion. To address this, HSP70 overexpression and downregulation (KD or KO) in U87 cells are necessary to T cells function test.